# A Survey of Robot Swarms’ Relative Localization Method

**DOI:** 10.3390/s22124424

**Published:** 2022-06-11

**Authors:** Siyuan Chen, Dong Yin, Yifeng Niu

**Affiliations:** College of Intelligence Science and Technology, National University of Defense Technology, Changsha 410073, China; chensiyuan12@nudt.edu.cn (S.C.); niuyifeng@nudt.edu.cn (Y.N.)

**Keywords:** robot swarms, localization technology, relative localization, microrobot

## Abstract

For robot swarm applications, accurate positioning is one of the most important requirements for avoiding collisions and keeping formations and cooperation between individuals. However, in some worst cases, the GNSS (Global Navigation Satellite System) signals are weak due to the crowd being in a swarm or blocked by a forest, mountains, and high buildings in the environment. Thus, relative localization is an indispensable way to provide position information for the swarm. In this paper, we review the status and development of relative localization. It is first assessed that relative localization to obtain spatio-temporal relationships between individuals is necessary to achieve the stable operation of the group. After analyzing typical relative localization systems and algorithms from the perspective of functionality and practicality, this paper concludes that the UWB-based (ultra wideband) system is suitable for the relative localization of robots in large-scale applications. Finally, after analyzing the current challenges in the field of fully distributed localization for robotic swarms, a complete mechanism encompassing the relative localization process and the relationship between local and global localization that can be a possible direction for future research is proposed.

## 1. Introduction

Biological swarms in nature realize complex group behaviors in the form of distributed control and self-organization under the interaction between adjacent individuals and the environment through simple autonomous decision-making rules and local sensing communication [1]. A biological swarm has the following characteristics:1.Collective robustnessThe biological swarm has a robust hierarchical structure that uses the interrelationship effect of the organizational framework to transform the structure to fill the gap when an individual failure causes a vacancy. Failure of a single individual does not significantly affect the biological swarm performance.2.Individual simplicityIndividuals within a swarm do not have a strong ability to accomplish the swarm’s tasks alone. Due to their simplicity, individuals accomplish complex tasks with each other through cooperative behavior with spatio-temporal relationships.3.ScalabilityWhen the number of individuals in a swarm increases, the control mechanism is still effective. The relative relationships within a swarm can be maintained steadily. The characteristics displayed by a swarm depend on the ability of individuals within the swarm to obtain the relative position of those around them.

Due to the limitations of endurance, sensing and load capacity, the robot’s ability to perform tasks alone is restricted. With the complex changes of task demands, people turn their attention to robot swarms, hoping that they can break through the limitations of individual and complete more complex tasks through group cooperation. Inspired by biological swarms, robot research has been developing in decision-making planning, communication networking, formation control, conflict resolution and other aspects [2]. Kushleyev et al. [3] used a VICON motion capture system to obtain the position information of 20 MAVs (micro aerial vehicles) for highly agile formation control. Liu et al. [4] used GPS (Global Positioning System) for the group positioning of 21 small fixed-wing UAVs (unmanned aerial vehicles) to complete formation control and collaborative recognition. Localization is the fundamental technology to achieve group robustness, cooperation and extensibility.

Localization can be divided into GNSS-based positioning and relative localization according to whether absolute location information can be obtained [5,6]. Biological swarms rely on sun and scene perception for self-positioning. Robots can obtain absolute position information provided by GNSS such as GPS, BeiDou, GLONASS, and Galileo. Relative localization is used to determine the relative position of an individual robot relative to other agents when absolute position information cannot be obtained [5]. Relative localization technology mainly involves measuring the distance/angle between individuals by sensors and then calculating the relative position coordinates of individuals relative to other units, which mainly includes sensor measurement methods and relative localization algorithms.

With the development of technology, the accuracy of GNSS-based positioning has been continuously improved. The localization precision of GNSS using RTK (real-time kinematic) technology reaches the centimeter level [7,8]. However, the GNSS positioning resolution will be reduced or the system will fail in dense buildings or mountainous jungles, or when individuals in dense groups block each other. Relative localization technology mainly solves the problem of robot positioning in complex environments and has become a research hotspot of robot motion control, multi-aircraft formation, cooperative detection and other applications [9,10,11].

Research on relative localization technology has been developed for decades [12]. At present, there are many kinds of measurement sensors and a variety of positioning algorithms, which can be used to solve the problem of relative positioning in complex indoor and outdoor environments for unmanned systems. It has been applied in UAVs, UGVs (unmanned ground vehicles), underwater robots, industrial robots and other fields, as shown in Table 1. However, with the urgent demand for large-scale robot applications in task scenarios (such as emergency and disaster relief, urban security, epidemic prevention and control, etc.), it is a major challenge to realize the relative localization of robots in a group dynamic environment.

According to our investigation, although necessary, there is currently a lack of reviews of the relative localization method of robot swarms. An existing survey in [22,23,24] focuses on a single location technique and does not provide a comprehensive review of other techniques. Shule et al. [22] focused on the multi-robot positioning technology of UWBs. Couturier and Akhloufi [23] introduced the absolute visual localization method of UAVs, and Motroni et al. [24] introduced the indoor vehicle localization technology of RFID. Some of the review articles in [25,26] provided an overview of the key technologies and application requirements of robot swarms but did not provide a detailed review of positioning technology. Wu et al. [16] commented on particular sensor positioning methods of underwater vehicles, which are not suitable for land and air due to the complex and changeable position of the robot and its operating environment. Kunhoth et al. [27] reviewed the positioning navigation system, but positioning as a part of the discussion is not detailed. As a detailed review of positioning, ref. [12,28,29] summarized the working principle of the existing positioning system and some challenges existing in the positioning system, most of which are aimed at a single robot, and few or no solutions are proposed. Therefore, it is crucial to summarize and discuss the localization method for robot swarms and determine the direction of further research. To further compare our work with existing literature in related fields, we show differences in Table 2.

Based on the EBSE (evidence-based software engineering) method in [30], this paper conducts systematic scanning and analyzes the development of relative positioning technology. The main contributions of this paper are as follows:This work examines and describes a number of features of relative localization systems that are appropriate for robot swarms.This work analyzes and evaluates typical positioning systems in terms of measurement frequency, positioning mode, and weight energy consumption, with the goal of meeting the application requirements of large-scale, high dynamic networking, and flexible mobility of robot swarms. According to our findings, UWB has the potential to be used in large-scale robotic swarm applications.This work considers the challenges that need to be met when the relative localization technology is applied to large-scale robot swarms.For a large number of robot swarms, a fully distributed relative localization method and its essential technologies are explored in this study, which can realize local and global localization.

The rest of this article is shown below. Section 2 discusses sensors and localization algorithms, and compares the advantages and disadvantages of localization technologies applicable to different scenarios. Section 3 describes the development of UWB positioning technology based on task requirements, and discusses the technology status quo. Section 4 discusses the challenges faced by UWB distributed relative positioning techniques when applied to large-scale robot swarms. Section 5 discusses future research directions for realizable distributed relative positioning mechanisms. Section 6 makes a brief conclusion to this paper.

## 2. Relative Localization Development Status

This section introduces and discusses the current development of relative localization technology. First, typical sensors such as Bluetooth, Wi-Fi, RFID (radio frequency identification), UWB, visible and infrared devices are introduced. At the same time, the basic principles of several classic positioning algorithms are introduced. After that, this paper compares and analyzes the advantages and disadvantages of various technologies in typical applications, and summarizes the relative localization technologies that can be applied to large-scale robot swarms.

### 2.1. Measurement System

We use sensors to get raw measurement information. Typical sensor devices that can be used for relative localization include Bluetooth, Wi-Fi, RFID, UWB, lidar, RGB camera, infrared camera, etc.

#### 2.1.1. Bluetooth

Bluetooth works in the 2.4 GHz ISM (Industrial Scientific Medical) band, with fast signal attenuation and weak penetration. When adjacent devices communicate, RSS (received signal strength) can be used to obtain distance information [31], and an antenna array can also be used to obtain angle information. The Bluetooth 5.0 signal range for positioning can reach 100 m, and the distance is calculated by modifying the signal intensity attenuation model in the environment, with accuracy to the meter level [32]. The angle accuracy of the obtained direction when using an antenna array is 8° [33]. Bluetooth can be used to locate mobile devices with slow motion speed [34].

#### 2.1.2. Wi-Fi

Wi-Fi works in the ISM band of 2.4 G/5 GHz, and the signal range of Wi-Fi 4 can reach 250 m [35]. Similar to Bluetooth, Wi-Fi uses RSS to obtain distance information or an antenna array to obtain angle information, which can be used for outdoor emergency search positioning and indoor mobile device localization. The distance accuracy is at the meter level [14], and the direction error is less than 9° [36].

#### 2.1.3. RFID

RFID has LF (Low Frequency, 125 KHz), HF (High Frequency, 13.56 MHz), UHF (Ultra High Frequency, 433 MHz, 860–960 MHz, 2.4 GHz) and SHF (Super High Frequency, 5.8 GHz) frequency bands [24]. The effective reading range of the UHF passive label is 10 m [18,37,38]. RFID of UHF can read the phase of the tag signal response and has a linear relationship with the distance between the detector and tag. The signal phase is integrated with RSS, and the measurement accuracy can reach the cm level, which can be used for indoor warehouse location tracking or UAV positioning tracking [37,38]. To improve the localization accuracy, Bernardini et al. [39] proposed a synthetic aperture radar (SAR) localization method for UHF-RFID tags by properly combining the phase data associated with a set of multiple paths, and the total length of the combined synthetic aperture increased, which in turn can improve the localization accuracy to approximately 4 cm.

#### 2.1.4. UWB

UWB devices operate in the 250–750 MHz, 3244–4742 MHz, and 5944–10,234 MHz frequency bands [40]. They transmit data signals using narrow nanosecond non-sinusoidal pulses with a bandwidth of 500 MHz or more. Their signals have an effective range of about 100 m and are highly penetrating and resistant to multipath [5]. The time of flight and thus the distance is solved by measuring the transmission of frames between two nodes with a measurement error in the centimeter level and a frequency up to 372 Hz [41]. It can be used for ranging and localization of indoor mobile devices and outdoor robots [42,43].

#### 2.1.5. Lidar

Lidar [44] uses 905 nm or 1550 nm light to scan the environment and detect a distance of about 200 m [45]. It uses the propagation time of light reaching the object and reflecting to calculate the point cloud information of the distance between it and the environment. The point cloud information is used to analyze the relative position of the device compared to the surrounding objects with a level of accuracy at the centimeter level. Lidar can be used to solve the problem of road recognition and obstacle avoidance in autonomous vehicle driving [46].

#### 2.1.6. RGB Camera

Similar to lidar, RGB cameras [47] are used to collect image information, extract features and construct scenes through algorithms to determine their positions from them. The accuracy of the position solution is within 20 cm. It is used for scene building and obstacle avoidance problems for UGVs and UAVs [48,49].

#### 2.1.7. Infrared Camera

Infrared cameras use 850 nm infrared light to illuminate objects and receive reflected images. A typical infrared camera indoor positioning system is a motion capture system [50], which uses an array of infrared cameras to track reflective marker points and calculate coordinate positions. A single camera has a detection range of about 10 m and can provide two-dimensional coordinates of reflective marker points in the field of view with sub-millimeter accuracy. The data provided by the motion capture system can be used as ground truth for algorithm verification. However, the system is large and time-consuming to install and calibrate and unusable when the reflective marker points are obscured.

#### 2.1.8. Summary of Measurement Methods

Table 3 combs and displays the sensor data used for positioning in this section. These measuring devices operate in cooperative modes that need to range each other and non-cooperative modes that require autonomous recognition by individuals. The system based on UWB has the characteristics of fast update frequency, long detection range, large location range and little influence by NLOS (non-line of sight). Positioning technology also adopts other sensors. However, ZigBee has large latency. Visible light communication is difficult to overcome NLOS. Ultrasonic distance is short. Geomagnetics are influenced by environmental factors. These sensors are not widely used for positioning, so they are not in the scope of this paper.

### 2.2. Location Algorithms

There are many kinds of relative localization algorithms [51], but the most typical ones are those based on RF (Radio Frequency) communication and optical signal.

#### 2.2.1. Positioning Algorithms Based on RF

RSS

RF-based communication technology can provide signal strength information [38], which is applied to devices with radio sensors [52]. RSS uses a model of the relationship between signal strength attenuation and distance to estimate the distance value between a tag and an anchor. The tag solves for its position using the distance value to each anchor and the position of each anchor. RSS is classically used in Bluetooth, Wi-Fi, and RFID-based positioning algorithms. The distance values between the tag and more than three anchors are required to calculate the planar coordinates, and more than four anchors are required for 3D coordinates. The localization accuracy is related to the ranging accuracy. The localization error of Bluetooth [32] and Wi-Fi [35] is usually at the meter level. Regarding UHF-RFID, by combining POA (phase of arrival) data [38], its localization error is at the centimeter level. The algorithm is computationally small and fast and can run on an embedded chip. However, this technique requires anchors with known deployment coordinate positions, and a large workload to correct the signal strength attenuation model. Furthermore, it is affected by environmental interference.

TOA (Time of Arrival)

Similar to RSS, TOA [53] calculates the distance value based on the time of flight of electromagnetic waves between devices and then solves the position coordinates using a distance-based localization algorithm. Suitable for UWB, the localization error is about 20 cm. The TOA has low computational complexity and is less affected by environmental interference. However, a tag needs to interrogate the anchors sequentially, which consumes some time.

AOA (Angle of Arrival)

AOA [54] obtains the angles at which the signal sent by the tag reaches anchors by using an array of antennas. We use these angles with the position of anchors to calculate the location of the tag. In Bluetooth and Wi-Fi systems, fingerprint maps are constructed to reduce errors to improve indoor positioning accuracy to the decimeter level [33,36]. The hardware structure and algorithm of the anchor for obtaining angle information are more complicated. It can only be used in known spaces and is highly influenced by environmental disturbances.

TDOA (Time Difference of Arrival)

TDOA [55] uses tags to send electromagnetic waves to anchors that have been time-synchronized. The anchors at different locations receive electromagnetic waves at different moments. The upper computer uses the time difference of these moments to calculate the tag position centrally. UWB can use this algorithm with a positioning error of about 20 cm. The positioning accuracy is limited by the time synchronization error of the anchors, as well as the environmental interference. The tag cannot calculate its own position and can only be obtained from the centralized calculation node.

#### 2.2.2. Positioning Algorithms Based on Optical Signals

SLAM (Simultaneous Localization and Mapping)

SLAM [56] helps robots to accomplish map building and localization in unknown environments using sensor information. This approach can be used for robots loaded with lidar or RGB cameras to obtain relative position information. Lidar SLAM [57] has centimeter-level localization accuracy, and vision SLAM [47] has a localization error of less than 20 cm. This algorithm consumes a lot of computational resources and is affected by the environment (e.g., light, rain, fog, etc.).

Multi-Camera Target Identification and Location Algorithm

The multi-camera target identification and localization algorithm determines the three-dimensional location of the target by capturing the two-dimensional position with a minimum of two cameras. The motion capture system uses an infrared camera array to solve the coordinates of the placed reflective marker points with sub-millimeter error. This algorithm centralizes the data from fixed nodes and consumes large computational resources. It is highly influenced by the environment, and the system cannot be used when the marker points are obscured.

To visually demonstrate the positioning algorithm introduced in this section, the measurement data required by the positioning algorithm, the basic process of measurement, required conditions and typical platforms are sorted out, as shown in Table 4.

### 2.3. Typical Positioning System

According to the positioning mode, this paper divides the positioning system into an “anchor-tag” mechanism and a full distribution mechanism.

#### 2.3.1. “Anchor-Tag” Mechanism

The mechanism has anchors with known positions pre-installed in the scene (whether the anchor moves or not). The tag initiates measurement communication with the anchor, and the position of the tag relative to the anchor is solved by an algorithm. Typical positioning systems using this mechanism are based on Bluetooth, Wi-Fi, RFID, UWB, infrared cameras, etc.

Bluetooth

Obreja and Vulpe [32] studied an indoor localization scheme based on Bluetooth beacon technology and an RSS fingerprinting method for indoor lightweight localization techniques. The scheme uses six Bluetooth beacons as anchors; more than 80 percent of the errors are within 1 m, and the rest are within 6 m. Wang et al. [33] designed a single-anchor positioning system based on angular information using an antenna array with direction and polarization information for positioning, with a median accuracy of 30 cm. Chen et al. [58] proposed an unsupervised indoor positioning system. The system combines data from iBeacons, Wi-Fi fingerprints and smart phone sensors to automatically establish a fingerprint database without any on-site survey. The average localization error is about 1.1 m in the steady state, and the maximum error is 2.77 m.

Wi-Fi

Rubina et al. [14] proposed a method to locate surviving devices using RSS of Wi-Fi. An aerial UAV carrying a Wi-Fi base station was used for emergency rescue to localize an area of 160,000 m ${}^{2}$ with meter-level accuracy. Kotaru et al. [59] proposed a Wi-Fi-based indoor localization system for locating human objects in indoor environments, providing a median accuracy of 40 cm for tracking tags and smartphones with Wi-Fi modules. The system uses access points with three antennas to create a virtual sensor array. It provides a higher accuracy AOA algorithm and performs position state estimation by fusing RSS information from each access point. Carvalho et al. [60] used machine learning technology to evaluate the faults of an indoor positioning system, and then proposed a fault-tolerant indoor positioning system [61]. The system uses the RSS set of Wi-Fi as input, and the RNN (recurrent neural network) determines the position of an agent according to the set. The system can distinguish momentary failure and permanent failure by a fault-tolerant mechanism.

RFID

Zhang et al. [37] proposed a UAV system using RFID in order to provide an accurate attitude to an indoor UAV. This system uses several readers with known locations to read the POA and RSS information fed by three UHF tags set on the UAV. Based on this information, distance values are calculated, and the position of the tags in the global coordinate system is tracked with a positioning error of approximately 0.04 m.

UWB

Chen et al. [62] optimized the UWB measurement method to solve the high-frequency positioning problem of mobile robots. The positioning refresh interval in the “Anchor-Tag” mode only needs 4.167 ms, and the 3D positioning error of UGV is within 20 cm. To reduce the errors generated by UWB devices subjected to multipath effects and NLOS, Liu et al. [63] proposed an effective framework for an integrated INS (inertial navigation system) and UWB positioning system for autonomous indoor mobile robots with a positioning error of about 20 cm. Li et al. [10] achieved 80 Hz 3D positioning of 6 micro UAVs based on the fusion of UWB and IMU, with an average error of 16 cm.

Infrared Camera

Motion capture systems locate a moving reflective marker ball by a fixed infrared camera array [50,64]. The data results are used as ground truth with an accuracy in the sub-millimeter range.

#### 2.3.2. Full Distribution Mechanism

In this mode, there is no limit to the anchors in the scene, and centralized calculation is not required. By accepting external information or active detection, the position of the agent relative to the map can be calculated, which can be called a complete distribution mode. Methods of measurement using this mode include UWB, lidar and RGB cameras.

UWB

Cao et al. [65] designed a fully distributed UWB relative localization scheme based on TDMA (time division multiple access) and a self-assembling network. It implements 12 nodes to construct a two-dimensional global map by relative localization. Under the condition of a 50 ms time slot, it takes about 30 time slots to complete one relative localization (positioning refresh interval 1.5 s). The positioning accuracy is about at the decimeter level under the condition that the points remain stationary.

Lidar

Lu et al. [57] proposed a lidar autonomous driving positioning framework based on learning in order to solve the problem of inaccurate manual modeling of autonomous driving positioning. It can directly process lidar point cloud data and accurately estimate vehicle position and direction, achieving centimeter-level positioning accuracy.

RGB Camera

Zhang et al. [47] aimed to tackle VIO’s vulnerability to poor light and featureless environments; thus a RGB camera was used to build a three-dimensional map matching algorithm based on conditional random field and VIO’s indoor positioning algorithm, achieving decimeter-level positioning accuracy.

#### 2.3.3. Summary

Figure 1 summarizes the above positioning systems, mainly showing the relationship between equipment, algorithms, positioning mechanisms and accuracy. Based on the analysis of group characteristics and the induction of Figure 1 and Table 3, a positioning system with decimeter-level accuracy, long detection distance, independence of prior knowledge and flexible deployment is suitable for robots. We found that Bluetooth and Wi-Fi are not very accurate, and it is difficult to establish a fingerprint database of current environment information. RFID’s effective working distance is short. Infrared cameras rely on fixed deployment. We preliminarily concluded that UWB-based, lidar and visual SLAM technology with high accuracy and full distribution may be suitable for robot swarms.

### 2.4. Analysis of Relative Localization Technology Matching with Robot Swarm Applications

In an emergency task scenario, a robot swarm should have the characteristics of the micro-miniaturization of the platform, low power consumption/lightweight load and limited energy, etc. The swarm has hierarchical relationship in communication and management, and the space-time relationship between individuals changes rapidly, so it must have the ability of mutual perception and collision avoidance [66]. Several characteristics of robot swarm relative positioning technologies are obtained by analysis as follows:Sensors with the Characteristics of Being Lightweight, Having Low Power Consumption, and Being Low-costThe sensors for relative measurement are lightweight, have low power consumption, and have low cost [67]. These features make the sensors easy to mount on the robot and stable to operate. Carrying lighter weight and lower power consumption sensors can reduce the load and consumption of robots. The lower cost facilitates robot cluster scaling.Fully Distributed Localization Mechanism with RobustnessThe relative localization mechanism should be adapted to the dynamic topology between nodes [68]. The number of node scales increases or decreases with task changes, scene changes, and confrontation conditions. Group nodes can still be positioned relatively under changes in topological structure.Obtaining Positioning Information in a Very Short TimeRelative localization, as the fundamental access control loop of navigation [69], enables robot movement to be completed following a plan. Swarms of drones acquire faster positioning information, enabling more responsive control and more demanding mission requirements.

In a crowded dynamic environment, three aspects need to be considered for the application of micro-robots in large groups. First, it is important to focus on the power consumption, sensing distance, weight, and cost of sensors. Second, the localization mechanism should be considered in terms of the localization mode and cooperation method. Third, it is important to pay attention to the measurement frequency and localization solving delay in the update frequency.

A single UWB node [62,70] weighs about 5 g [71] and has an operating voltage of 3.3 V and a current of 130 mA [41]. Each node costs tens of dollars. As regards cooperative positioning, the node measurement can be selected by the host computer and RF chip. The measurement frequency can reach 372 Hz [41], which can adapt to the motion loop of a 50 Hz control loop platform [72].

RGB cameras [47,50] weigh about 100 g and have an average power of 0.36 w [73]. Each camera costs about several hundred dollars and can provide services for the self-positioning of robots without the need to cooperate with other robots. Although the visible range is all detected, the range distance for building maps is small. Map construction requires datasets for training, a long pre-learning time, and the need to process environmental information. The more complex the external environment, the higher the algorithm delay.

Lidar [46] weighs nearly 1 kg, has an average power of 10 W [45], and costs several thousand to several tens of thousands of dollars individually. Similar to RGB cameras, robots can use point cloud information from lidar for self-localization. Although lidar can detect objects up to 200 m, the point cloud information is too sparse at long distances, and the sensing distance is much smaller than the detection distance. The measurement frequency is affected by the hardware scanning speed, as fast as 20 Hz, and is susceptible to smoke obscuration.

We summarized UWB, RGB camera and lidar in terms of equipment parameters and capabilities, as shown in Table 5. Compared with RGB cameras and lidar, UWB has low power consumption, a large sensing distance, is lightweight, low-cost, has a variable positioning mode, and has a high measurement frequency, making it more ideal for fast-moving robot platforms in dynamic settings.

## 3. Development of Relative Localization Technology Based on UWB

According to our literature review, Fontana and Gunderson [74] first proposed that UWB can be used for “anchor-tag mode” positioning systems. They evaluated the localization accuracy of UWB in the presence of multipath propagation in the shipboard environment, and the localization error reached the decimeter level for fixed points in the cabin. Krishnan et al. [75] retrofitted UWB tags to a movable robot for indoor robot localization and navigation to construct a TDOA algorithm-based localization and navigation system. The system has a planar localization error of less than 25 cm, but there are distance outliers. Cheok et al. [76] designed a UGV relative localization system. The system contains movable-position self-calibrating anchors, the distance outliers are handled by a fuzzy neighborhood tracking filtering technique, the estimated results are compared with the video, and the results are generally consistent.

### 3.1. Indoor UAV Application

With the extensive application of UAVs, the demand for positioning is increasing. Aiming at the indoor application environment, researchers have studied the methods of improving the relative localization accuracy and frequency of a single UAV and multiple UAVs. Tiemann et al. [71] mounted UWB tags to MAVs and deployed UWB anchors fixed indoors. They used the anchor-tag mode to achieve indoor 3D relative localization. They counted 8960 localization results, with 95 percent of their localization errors within 20 cm and 99 percent within 30 cm. These positioning results still have a small number of larger error values and are updated slowly, which is lower than the motion control loop requirements. Guo et al. [77] used UWB to realize that the relative localization error of a single UAV in the plane was within 20 cm, and the relative localization frequency was increased to 40 Hz, but there was no height positioning information. Nguyen et al. [78] used the TOA algorithm to provide positioning information of 20 Hz for 4 indoor UAVs, with plane error within 20 cm and height error within 40 cm. Tiemann et al. set up a TDOA positioning system serving multiple users [79,80] and distributed the positioning information of three UAVs using centralized computing and WLAN (Wireless Local Area Network) [13]. The position update rate was greater than 40 Hz, the plain error was within 20 cm, and the height error was within 40 cm.

### 3.2. Outdoor UAV Applications

UAVs are mostly used in outdoor environments, and many studies have researched relative localization in outdoor environments. Steup et al. [81] set UWB anchors in an outdoor environment to deal with the problem of GPS signal loss, providing UAVs with a positioning system with an accuracy of about 0.2 m. Lazzari et al. [43] deployed a UWB relative localization system containing four movable anchors on a vehicle to meet the tracking and control of UAVs in the following convoys. The system had a horizontal error of 30 cm, an altitude error of 50 cm, and a positioning refresh interval of 6.5 ms to 12.5 ms. Cao et al. [70] addressed the challenge of close formation of multiple fixed-wing UAVs in lead-follow mode by loading a UWB positioning system with three anchors on a pilot UAV. The system uses GPS to guide the aircraft to approach each other at larger initial distances and turns on the UWB device at less than 100 m to maintain formation flight with a horizontal positioning error of 60 cm or less.

### 3.3. Fusion

To improve the accuracy of the UWB positioning system, Hol et al. [82] proposed a tracking system that integrates UWB and low-cost IMU, which can estimate the position and direction of sensor units under multipath effect and NLOS conditions to detect outliers in processing ranging. As the fusion method is convenient, a large part of the UWB positioning system of UAVs adopts technology fusion to improve the accuracy of positioning information acquisition. Li et al. [10] improved the fusion algorithm of UWB and IMU, and the average positioning error decreased from 30cm to 16cm. Xu et al. [83] used UWB to initialize the relative positions of multiple UAVs, calculated the position information by the visual-inertial odometer, and continuously corrected it by UWB. Qi et al. [11] obtained relative location information by using UWB and IMU fusion information and then solved geographic location information by GPS. Nguyen et al. [84], to solve the precise positioning challenge of UAV landing on the unmanned vehicle, used UWB positioning to guide the UAV to approach the unmanned vehicle, and then integrated vision and UWB information to accurately guide UAV landing on the roof of 1.5 × 1 m, with a positioning accuracy of 20 cm. Cao and Beltrame [85] used UWB to assist in correcting the visual-inertial odometer drift of a single robot.

### 3.4. Improvements Based on UWB

With the research of multi-sensor fusion state estimation, the research of UWB measurement system and sensor characteristics is also carried out simultaneously. To improve the measurement accuracy of the positioning chip, Sidorenko et al. [86] proposed a clock drift correction method without additional equipment, reduced the TOA ranging error of UWB to 1.5 cm, and further extended the clock drift correction method from TOA to TDOA [87]. On the premise of ensuring the high positioning frequency of TDOA, it was found to have the same accuracy as TOA. Chen et al. [88] proposed an improved method of SS-TWR based on UWB to solve the problem of the slow update frequency of TOA measurements, which improved the ranging frequency to more than 200 Hz under the condition of constant accuracy.

### 3.5. Summary

A summary of the above UWB positioning systems is givenin Table 6. UWB has been widely applied in the field of UAV positioning, mainly focusing on the anchor-tag mode. The method of solving the position of the moving tags on a coordinate system with known anchors position has a positioning error of 20 cm on average. However, existing studies using the anchor-tag model can only address a portion of the application challenges for robot swarms in real-world settings. For large-scale groups, the application relationship is changing, and the node model with roles is not applicable, so we will discuss the full distribution mode.

## 4. Challenges of UWB-Based Distributed Relative Localization Technology

Robots need to recognize which vehicles are around them, in which direction, and how far away they are in both indoor and outdoor environments. The robot has no pre-positioned anchors to rely on in a dynamic environment. The localization mode of anchor-tag cannot satisfy the swarm application research, and the distributed relative localization oriented to fully peer nodes must be explored. As a result, the robot’s position information can only be solved using UWB ranging information. Due to the pulse communication characteristics of UWB [89], it is difficult to use FDMA (frequency division multiple access) and CDMA (code division multiple access) in a single channel. When a single node sends a message, the channel it is on is occupied by it, and if the rest of the nodes send messages at this time, it will cause communication congestion. The growth curve of total ranging interaction time $T_{s}$ becomes fairly steep as the number of robots *n* increases. In a completely connected manner, the robot conducts one-to-one ranging interactions *Q* times, with the ranging time being $t_{r}$, while the ranging interaction time is
$$T_{s}=t_{r}Q=t_{r}C\left(2,n\right)=t_{r}\frac{n^{2}-n}{2}$$
where C(2,n) is the combination number formula. The time spent obtaining position data increases until the maximum tolerable delay time for robot movement is reached, at which point the number of robots in the swarm meets its maximum capacity. If the number of robots continues to rise, it will be necessary to manage them in groups. That is, the number of robots in each group does not exceed the upper limit, and local maps can be built. Adjacent groups can carry out coordinate transformation through multiple connected points to build a global map. As an example, in Figure 2, we have created a robot swarm with three groups. The swarm has a global map, while each group has its own local map. The main challenges are as follows:

### 4.1. When Robots Join and Exit, How Do They Dynamically and Swiftly Construct Groups and Pick Group Members?

Currently, there is a lack of a positioning mechanism, i.e., a complete set of processes including “initialization, local measurement, decomposition, local positioning information update, and global positioning information update”. The literature [90] investigated the 3D spatial localization problem for a set of UAVs with only distance measurement interactions, established constraints on the minimum number of distance measurements required for localization, and performed coordinate updates. Costa et al. [91] proposed a distributed localization algorithm that includes global updates to conduct localization information to all nodes in order to solve the global map update problem. The localization method in the literature [65] considers node changes with global map updates in two dimensions, but the update frequency is slow and can only localize stationary targets.

### 4.2. A Method for Determining Relative Location in Real Time Based on Channel Capacity

Swarms require collective robustness [1], and the loss of a single individual does not affect overall performance. The anchor-tag mode that is widely applied at present does not meet the needs of the group. The literature [92] has proposed a completely distributed dynamic positioning method, in which multiple robots build adaptive coordinate systems only through relative distance information. However, the method was developed in a simulation environment without taking into account the UWB channel capacity, and the two-dimensional positioning method is challenging to expand to three-dimensional space. Other literature [93] explores the relative localization method of multiple UAVs with a complete distribution mode, realizing relative localization in three-dimensional space. However, UAV position estimation requires additional prior position and velocity information, and the results will also face divergence challenges. A further study [83] only uses UWB ranging information for the relative localization of multiple UAVs without prior information, but it is only used during system initialization without a continuous coordinate update.

### 4.3. To Accommodate More Robots, the Group Adopts an Excellent Network Structure

In the existing studies, there are 6 or 7 robots based on UWB relative localization [11,83]. However, as the number of robots increases, the distance message interaction time increases, and the local group size faces challenges. Cao et al. [94] used TDMA technology to avoid communication congestion and expand the number of nodes in a fully distributed mode. However, they did not consider the errors arising from the dynamic node motion. For mobile robots, the errors caused by motion will become unacceptable as the number of nodes increases. As far as the author knows, there is no relevant study on the size of relative localization agreement group members.

## 5. Future Study on Huge Robot Swarms’ Decentralized Relative Localization

We propose an implementable solution to the challenges of applying relative localization technology to large-scale robotic swarms.

### 5.1. Relative Localization Mechanism

The positioning mechanism of robot swarms should be built from individual nodes, which have modules of initialization, adjacency inquiry, local positioning and global positioning, and can be executed sequentially as shown on the left side of Figure 3. A robotic swarm is seen in the upper right corner. A robot in the swarm is represented by each node. The swarm is divided into i groups, and coordinate information is passed between them via connected points connections.

### 5.2. Initialization

In order to prevent communication congestion, we need to allocate channel resources reasonably and adopt polling multiple access technology. The polling mechanism can either allocate time slots for nodes using TDMA synchronously [65] or asynchronously using token passing [95], and the mechanism used is determined at the initialization time. Nodes use extensive communication broadcasts to initialize the whole system. The nodes that receive the communication broadcast can know the neighboring nodes and thus establish the neighboring node table.

### 5.3. Ask for Neighbors

To adapt to the dynamic changes in the system, the adjacency node list needs to be updated periodically according to the rules.

### 5.4. Local Localization

#### 5.4.1. Select Measurable Neighbors

UWB has the best measurement range, and the positioning accuracy of nodes beyond the range is reduced [96]. We select the nodes in the best measurement range from the adjacency node list to generate a measurable neighbor node list.

#### 5.4.2. Measurement Process

The nodes are polled for measurements based on the initialized rules. The measurement method used needs to balance time with packet loss during measurement.

#### 5.4.3. Relative Position Calculation

The nodes are distributed to solve the relative position by distance measurement information. To reduce the error caused by relative moving position in the measurement process, methods such as shortening the distance measurement time to shorten the time slot length [88], a minimum information positioning algorithm suitable for a UWB communication mechanism [90], and fusion with IMU and other sensors to reduce the error [10] can be explored.

### 5.5. Global Localization

It is necessary to construct some connection points between locals to complete coordinate transformation between local and global graphic calculation. Local coordinate systems are unified into global coordinate systems through mutual transformation. In this process, overlapping connection points must exist between adjacent locals [97]. Therefore, one must to establish join point selection methods. To improve the efficiency of global positioning, all nodes need to be traversed quickly. This traversal method is suggested to refer to the flooding mechanism of network routing [98].

## 6. Conclusions

In this paper, the relative positioning technologies for large robot swarm are analyzed and summarized. Firstly, the existing measurement systems of different sensors and typical positioning algorithms are discussed. After that, two different localization systems are examined: anchor-tag and complete distribution. According to our examination, se summarize several characteristics of relative localization technologies suitable for robot swarms. This paper analyses the typical positioning system from the measurement frequency, positioning mode, weight energy consumption and other aspects. Based on our analysis, we conclude that UWB has the potential for large-scale robotic swarm applications.

In particular, we sort out and summarize the current situation of indoor and outdoor applications and the integration and improvement of UWB relative positioning technology according to development. We then analyze the challenges that will be encountered with relative positioning technology as the number of robot swarms increases. Finally, a more detailed fully distributed relative localization mechanism that can be implemented and its key techniques are discussed for swarms with a large number of robots in grouped networks, which can achieve local and global localization and provide some promising directions for future research.

## Figures and Tables

**Figure 1 sensors-22-04424-f001:**
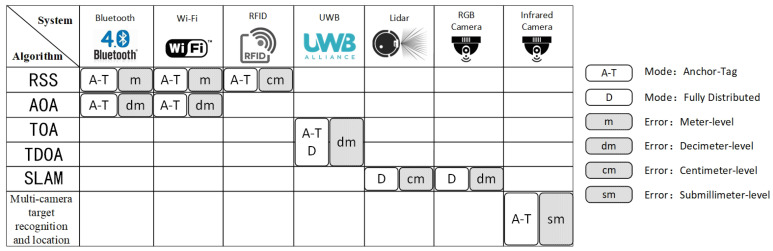
We show the different systems and the algorithms that can be used, as well as the application patterns and error ranges under this algorithm.

**Figure 2 sensors-22-04424-f002:**
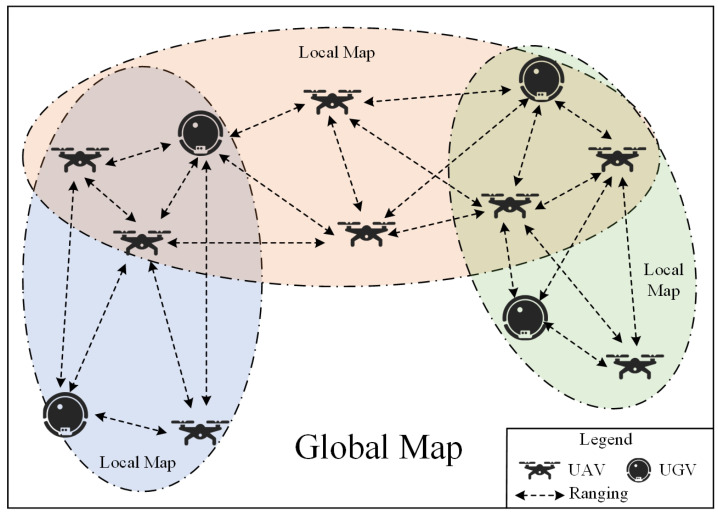
This diagram shows the relationship between groups and a swarm. Groups have local maps, and the swarm have global maps. There are connected points between adjacent groups, which will help them coordinate conversion.

**Figure 3 sensors-22-04424-f003:**
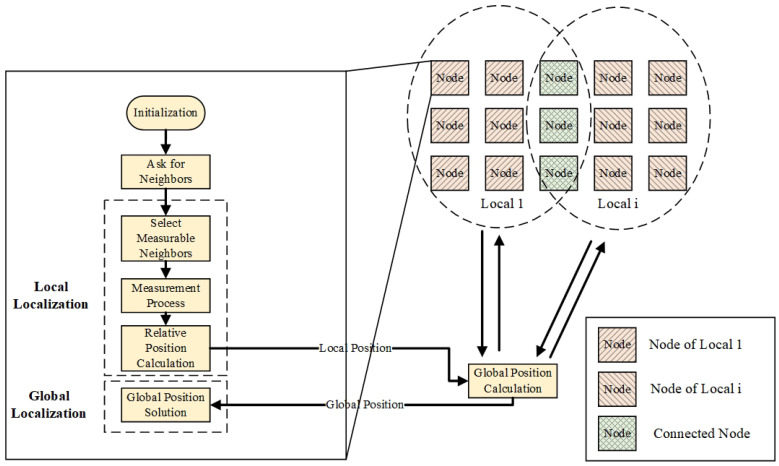
Schematic diagram of a large scale robot swarm localization mechanism.

**Table 1 sensors-22-04424-t001:** Summary of unmanned platform applications.

Scenario	Date	Platform	Positioning Method	Precision	Literature
Quick Inventory of Goods in the Warehouse	2017	Multiple UAVs	UWB/TDOA	Decimeter-level	[13]
Emergency Search and Positioning Rescue	2017	Multiple UAVs	Wi-Fi/RSS	Meter-level	[14]
Foraging Task	2017	Robots	Odometry+GPS	-	[15]
Underwater Operation	2019	Underwater Robots	-	-	[16]
Carrying, Welding, Spraying, or Other Industrial Applications	2020	Industrial Robot	-	Submillimeter-level	[17]
Warehouse Inventory Management	2021	UAV	UHF-RFID	Decimeter-level	[18]
Port Unmanned Cargo	2021	UGV	Laser SLAM	Centimeter-level	[19]
Hand-guiding Precise Assembly Operations	2022	Industrial Robot	-	Submillimeter-level	[20]
High Voltage Transmission Line Inspection	2022	UAV	Laser SLAM	Centimeter-level	[21]

**Table 2 sensors-22-04424-t002:** Comparison with the existing literature.

Literature	Date	Robot Swarm Application	Multiple Localization Mechanism	Evaluation Framework for Robot Swarms	Existing Challenges	Solution
Wu et al. [16]	2019	No	Yes	No	No	No
Tahir et al. [26]	2019	Yes	No	Yes	No	No
Zafari et al. [12]	2019	No	Yes	No	Yes	Yes
Shule et al. [22]	2020	Yes	No	No	No	No
Coppola et al. [25]	2020	Yes	No	Yes	Yes	Yes
Kunhoth et al. [27]	2020	No	Yes	No	No	No
Couturier and Akhloufi [23]	2021	No	No	No	Yes	Yes
Motroni et al. [24]	2021	No	No	No	Yes	No
Yang and Yang [29]	2021	No	Yes	No	Yes	Yes
Yuan et al. [28]	2021	No	Yes	No	Yes	Yes
Our work	2022	Yes	Yes	Yes	Yes	Yes

**Table 3 sensors-22-04424-t003:** Typical data comparison of sensors between different measurement systems.

Sensor	Frequency	Detection Range	Accuracy of Measurement	Requirement	NLOS Effect	Working Mode
Bluetooth [31,32,33,34]	Up to 5 Hz	≤100 m	Meter-level	Based-anchor	Medium	Cooperative
Wi-Fi [14,35,36]	Up to 10 Hz	≤250 m	Meter-level	Based-anchor	Medium	Cooperative
RFID [18,24,37,38]	Up to 50 Hz	≤10 m	Centimeter-level	Based-anchor	Medium	Cooperative
UWB [5,40,41,42,43]	Up to 372 Hz	≤100 m	Centimeter-level	Based-anchor or non-anchor	Small	Cooperative
Lidar [44,45,46]	-	≤200 m	Centimeter-level	-	Big	Non-cooperative
RGB Camera [47,48,49]	-	-	Decimeter-level	-	Big	Non-cooperative
Infrared Camera [50]	-	≤10 m	Submillimeter-level	Camera Array	Big	Cooperative

**Table 4 sensors-22-04424-t004:** Data types, measurement process, required conditions, and typical platform for various positioning algorithms.

Algorithm	Type of Data Measured	Schematic Diagram of Measurement Process	Required Conditions	Typical Platforms
RSS [32,35,38,52]	Signal strength	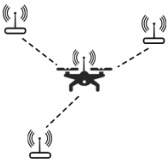	In order to improve the accuracy of fingerprint map construction	Bluetooth, Wi-Fi, RFID
TOA [53]	The transmission time of electromagnetic waves between terminals	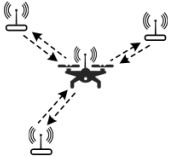	Single point multiple measurements	UWB
AOA [33,36,54]	The array antenna receives electromagnetic signals out of phase	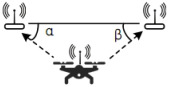	Array antenna, phase difference solution, angle algorithm	Bluetooth, Wi-Fi
TDOA [55]	The time difference between the electromagnetic wave of the measured terminal and different terminals	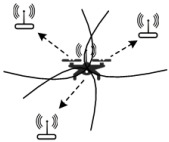	Accurate time synchronization on the terminal	UWB
SLAM [47,56,57]	Image, laser ranging information	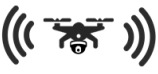	The visual must be visible	Lidar, RGB Camera
Multi-Camera Target Recognition and Location Algorithm	Infrared image	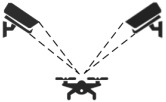	The visual must be visible	Motion capture system

**Table 5 sensors-22-04424-t005:** Typical data comparison table of UWB, RGB cameras and lidar.

Classification	UWB	RGB Camera	Lidar
Power consumption:	Low	Low	High
Perceived distance:	100 m	-	-
Weight:	12 g	100 g	925 g
Single price:	Dozens of dollars	A few thousand dollars	Tens of thousands of dollars
Positioning mode:	Anchor-Tag or Distributed	Distributed	Distributed
Method of cooperation:	Cooperation	Non-cooperative	Non-cooperative
Environmental impact:	smaller	Data sets need to be learned	Affected by smog
Measurement frequency:	372 Hz	90 FPS	20 Hz
Positioning delay:	Lower	Environment complexity correlation	Lower

**Table 6 sensors-22-04424-t006:** UWB-based relative localization technology statistics.

Existing Literature	Synchronicity	Number of Nodes	Positioning Mode	Precision	Sensor
Fontana and Gunderson [74]	Asynchronous	4 Anchors, 1 Tag	A-T *	Meter-level	UWB
Krishnan et al. [75]	Asynchronous	4 Anchors, 1 Tag	A-T	Decimeter-level	UWB
Cheok et al. [76]	Synchronous	4 Anchors, 1 Tag	A-T	-	UWB
Tiemann et al. [71]	Asynchronous	8 Anchors, 1 Tag	A-T	Decimeter-level	UWB
Guo et al. [77]	Asynchronous	4 Anchors, 1 Tag	A-T	Decimeter-level	UWB
Nguyen et al. [78]	Asynchronous	4 Anchors, 4 Tags	A-T	Decimeter-level	UWB
Tiemann et al. [79,80]	Synchronous	8 Anchors, 3 Tags	A-T	Decimeter-level	UWB
Lazzari et al. [43]	Asynchronous	4 Anchors, 1 Tag	A-T	Decimeter-level	UWB
Cao et al. [70]	Asynchronous	3 Anchors, 1 Tag	A-T	Decimeter-level	UWB
Hol et al. [82]	Asynchronous	6 Anchors, 1 Tag	A-T	Decimeter-level	UWB+ IMU
Li et al. [10]	Asynchronous	6 Anchors, 1 Tag	A-T	Decimeter-level	UWB + IMU
Xu et al. [83]	Synchronous	5 Nodes	Distributed	Centimeter-level	UWB + IMU + Camera
Qi et al. [11]	Synchronous	7 Nodes	Distributed	Decimeter-level	UWB + IMU + GPS
Nguyen et al. [84]	Asynchronous	4 Anchors, 1 Tag	A-T	Decimeter-level	UWB + Camera
Cao and Beltrame [85]	Asynchronous	1 Anchor, 1 Tag	A-T	Centimeter-level	UWB + IMU + Camera
Sidorenko et al. [86,87]	Asynchronous	2 Nodes	-	Centimeter-level	UWB
Chen et al. [88]	Asynchronous	2 Nodes	-	Centimeter-level	UWB

* The “A-T” in the table represents the anchor-tag mode.

## Data Availability

Not applicable.

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
