# Peer review of "A Survey of Robot Swarms’ Relative Localization Method"

_sensors, 2022, doi:10.3390/s22124424_

Round 1
Reviewer 1 Report
The writing and presentation quality of this article is good. A very precise and clear idea about this work is presented in both Abstract and conclusions. In this paper, the authors analyzed the main technologies and methodologies of relative localization. The authors also discussed current challenges in the field of 10fully distributed localization for robotic swarms. Finally, the trend of relative localization technology is forecasted in this study. I don’t have any specific comments; however, I have noticed a few issues which must be addressed to improve quality and develop more interest in readers:
- In the abstract, what is GNSS, UWB? Authors should carefully check each abbreviation and it must be defined at the first place of appearance for better readability.
- In Table 1, the authors should add more relevant studies.
- It is suggested to add a Table and compare your work with relevant reviews or surveys. Readers can fastly understand the research contributions of this study and previous works.
- Table 2 does not require any reference? I think these parameters have been reported in previous studies.
- Authors should revise tables and provide sufficient information and discussion of each reference study.
- Authors should add more discussion from related studies in the subsections of 2.1.
- Insufficient reference literature is provided in subsections of 2.2 and 2.3. Authors should add more reference literature.
- Authors have missed several interesting studies reported on indoor and outdoor UAV applications. Specifically, there is a complete lack of information provided for outdoor UAV applications.
- Authors have placed Figure 2 without complete analysis, benefits compared to other localization mechanisms, and how it can be helpful for the relevant research peers.
- Conclusion section must be revised. Authors should shed some light on possible research directions for future works which might be helpful for the relevant research fraternity.
- Authors should revise abbreviations as they did not define several acronyms which can create difficulty for its readers.
- Reference section is updated with very good relevant studies. I have found several interesting articles.
Reviewer 2 Report
-The paper although interesting, is related to a topic that has already been thoroughly studied and investigated. The standardization of this class of protocols is already very advanced and at least some indications should be provided in the text before acceptance.
-What is the difference between this survey and the existing ones?
-What is the major contribution of the work? Are the review, classification, and discussion of recent works in the area?
-Little is described or explained about figures and tables in the text. The authors should keep in mind that this manuscript is a survey, where the authors must target instructing the readers and clearly describing concepts and arguments.
-Important references are missing and should be added, such as: An Adaptive Energy Saving Algorithm for an RSSI-Based Localization System in Mobile Radio Sensors; Towards a smart fault tolerant indoor localization system through recurrent neural networks; Unsupervised Indoor Localization Based on Smartphone Sensors, iBeacon and Wi-Fi; Exploiting the use of machine learning in two different sensor network architectures for indoor localization.
-It lacked a section to discuss the current research challenge, as well as future topics to guide further research
Reviewer 3 Report
The authors have done interesting work that involves a systematic review of the literature with the aim of finding robot swarms’ relative localization. For the methodology of investigation of fufuros works, I suggest to carry out an approach based on the method of [1]. In this work, the authors make a systematic scan of the literature in several scientific bases, in addition, it is possible to find an appropriate methodology in cases of duplication and insert cases of exclusion of articles.
Regarding the works found, I believe that other methodologies could have been added to the work, such as the use of odometry and GPS to locate robots [2]. In addition, authors should use comment the work of [3] that uses positioning accuracy reliability analysis for industrial robots and comment the paper [4] that deals with a robot based indoor positioning of UHF-RFID tags, a study case of the SAR method with multiple trajectories. Another paper that is important is [5] where authors proposed an approach to robust INS/UWB integrated positioning for autonomous indoor mobile robots and comment the paper [6] where authors proposed precise positioning of collaborative robotic manipulators using hand-guiding. Finally, I recommend reading the articles presented in the Survey [7] to verify the similarities and dissimilarities found in each of them, that is, which articles were repeated and if the study presented here as a survey is not redundant in relation to the Survey in [7].
1. KITCHENHAM, Barbara et al. Systematic literature reviews in software engineering–a systematic literature review. Information and software technology, v. 51, n. 1, p. 7-15, 2009.
2. LIMA, D. A., et al. "Coordination, Synchronization and Localization Investigations in a Parallel Intelligent Robot Cellular Automata Model that Performs Foraging Task." ICAART (2) 2017 (2017): 355-363. 3. WU, Jinhui et al. A moment approach to positioning accuracy reliability analysis for industrial robots. IEEE Transactions on Reliability, v. 69, n. 2, p. 699-714, 2019. 4. BERNARDINI, Fabio et al. Robot-based indoor positioning of UHF-RFID tags: The SAR method with multiple trajectories. IEEE Transactions on Instrumentation and Measurement, v. 70, p. 1-15, 2020. 5. LIU, Jianfeng et al. An approach to robust INS/UWB integrated positioning for autonomous indoor mobile robots. Sensors, v. 19, n. 4, p. 950, 2019. 6. SAFEEA, Mohammad; NETO, Pedro. Precise positioning of collaborative robotic manipulators using hand-guiding. The International Journal of Advanced Manufacturing Technology, p. 1-12, 2022. 7. WU, Yinghao et al. Survey of underwater robot positioning navigation. Applied Ocean Research, v. 90, p. 101845, 2019.
Round 2
Reviewer 2 Report
authors responded appropriately